# Electrical Circuits Simulator in Null-Flux Electrodynamic Suspension Analysis

**Thaís N. França, Hengda Li, Hanlin Zhu, Hongfu Shi, Le Liang and Zigang Deng \***

Applied Superconductivity Laboratory, State Key Laboratory of Rail Transit Vehicle System, Southwest Jiaotong University, Chengdu 610031, China
* Correspondence: deng@swjtu.cn

**Abstract:** This paper employed an electrical circuit simulator to investigate an electrodynamic suspension system (EDS) for passenger rail transport applications. Focusing on a null-flux suspension system utilizing figure-eight-shaped coils (8-shaped coils), the aim was to characterize the three primary electromagnetic forces generated in an EDS and to compare the findings with existing literatures. The dynamic circuit theory (DCT) approach was utilized to model the system as an electrical circuit with lumped parameters, and mutual inductance values between the superconducting (SC) coil and the upper and lower loops of the 8-shaped coil were calculated and inputted into the simulator. The results were compared with experimental data obtained from the Yamanashi test track. The comparison demonstrated close alignment between the theoretical expectations and the obtained experimental curves, validating the accuracy of the proposed model. The study highlights the advantages of this new approach, including faster computation times and efficient implementation of modifications. Overall, this work contributes to the ongoing development and optimization of null-flux suspension Maglev systems.

**Keywords:** magnetic levitation; electrodynamic suspension; rail transport; figure-eight-shaped coil; null-flux suspension; superconducting coil





## 1. Introduction

The contactless nature of magnetic levitation (Maglev) trains offers increased speed and efficiency in transportation systems [1]. Currently, three magnetic levitation techniques are employed for Maglev transportation: high-temperature superconducting (HTS) flux pinning suspension, electromagnetic suspension (EMS), and electrodynamic suspension (EDS) [2]. HTS relies on the non-ideal diamagnetism and flux-pinning properties of the type-II superconductors to achieve stable electromagnetic forces between permanent magnet guideways and superconductors [3,4]. EMS utilizes the interaction between ferromagnetic materials and electromagnets, requiring a closed control loop, due to the unstable nature of the attractive force [5,6]. EDS, the focus of this work, is based on the induced currents forming in conductive materials when exposed to a variable magnetic field, generating repulsive forces [7,8]. Superconducting EDS is suitable for high-speed trains that lack commercial operation vehicles, with a Japanese prototype which is expected to launch in 2027 [9].

High-speed trains have proven to be a viable option for distances up to 1000 km, offering advantages, such as time and cost savings, travel comfort, and reliability, compared to other modes of transport [10]. Due to their contactless nature, Maglev trains distribute the vehicle's load along its entire length, reducing track stress and allowing higher speeds [11]. This results in a lighter, less expensive infrastructure with reduced maintenance requirements [12]. The U-shaped guideway structure enhances safety, as derailments are highly unlikely [13]. Additionally, the absence of traditional rotary engine components leads to a lighter vehicle with a reduced cross-sectional area, contributing to a more compact design and increased energy efficiency [14,15].

The calculation of magnetic forces in a Maglev system has peculiarities from a linear machine [16]. Radial forces between the stator and rotor provide lift for the vehicle, with spatial harmonics affecting the performance of the EDS system [17]. Transient effects of the temporal–spatial dependence of magnetic forces impact component vibration, noise, vehicle motion, and ride quality [18]. Various approaches are used to calculate these forces, including numerical models based on harmonic analysis via Fourier transform [19] or on finite element methods [20,21]. However, these methods often require significant computation time. Dynamic circuit theory (DCT) offers a more efficient alternative, considering the EDS system in terms of space and time-dependent circuit parameters governed by differential equations [22,23]. This method supports edge and end effects and is well-suited for dynamic and transient analysis and computer simulation.

This research was motivated by the need to develop a time-efficient method for calculating the forces generated by a null-flux EDS system, eliminating the lengthy computational processes required by finite element methods. This approach, which has not been explored in previous studies, aimed to simplify the research process involved in investigating electrodynamic levitation phenomena. By using this approach, researchers would no longer be burdened with writing extensive algorithms or relying on computationally demanding methods. Another motivation was to provide a straightforward way to study the non-cross-connected null-flux system. Implementing the system in the simulator could solve induced currents and generate forces quickly. Additionally, the research aimed to facilitate the transition to analyzing cross-connected systems by connecting conductive wires. This flexibility would facilitate the introduction of other electrical components and modifications to the electrical circuit, enabling further exploration and experimentation.

This paper presents a numerical calculation model using an electrical circuit simulator to analyze the interaction of superconducting (SC) coils and 8-shaped coils, based on DCT and Yamanashi test line data [24,25]. DCT has been used to calculate forces without utilizing d-q, or Fourier transforms. This method was used by He et al. (1993, 1994, 1995) [22]. However, the approach used in our study differs because our approach uses an electric circuit simulator, which eliminates the need to write a detailed algorithm line by line. Instead, it focuses solely on calculating mutual inductances and its derivatives. This streamlined approach can save time and effort while still producing accurate results. The dynamic circuit model was verified using the experimental data, and the electromagnetic force's properties were examined. The paper is organized as follows: Section 2 explores the fundamental principles and mathematical model underpinning an EDS system's operation; Section 2.1 presents a comparative analysis of other studies of EDS systems worldwide; Section 2.2 characterizes the basic principles of an EDS system; Section 2.3 establishes the mathematical model of a single-sided EDS system, focusing on the calculation and implementation of the mutual inductance in the simulator; Section 2.4 provides an overview of the computational algorithm implemented; Section 2.5 presents the calculation of the mutual inductance coupling between an SC coil and an 8-shaped coil; Section 2.6 details the implementation of the algorithm in the circuit simulator; Section 2.7 features the calculated electromagnetic forces; Section 2.8 validates the model through the use of experimental data; Section 2.9 discusses potential extrapolation for a more robust circuit; and Section 3 discusses the advantages and limitations of the method, and suggests future research to improve the system's performance.

## 2. Unveiling the Inner Workings: Exploring the Principles and Model of an EDS System

In this section, an in-depth exploration of the fundamental principles and mathematical model that underlie the operation of an EDS system is carried out. The implementation of the computational algorithm, which serves as the backbone of the analysis, is explored, shedding light on the system's inner workings. This comprehensive investigation seeks a deeper understanding of the underlying principles and model that govern an EDS system, paving the way for future advances and refinements in this field.

## 2.1. The Research Gap

EDS is studied and examined using a variety of techniques and frameworks. These techniques give researchers a variety of instruments and approaches to research and examine EDS systems, each having its benefits and considerations, depending on the particular goals and complexities of the system under study.

Analytical models based on theoretical concepts, such as electromagnetic theory and mechanical dynamics, are created to research particular facets of EDS. For better system performance and to acquire insights into system behavior, these models frequently simplify mathematical equations and presumptions. Physical models in scientific research often need more than oversimplification. As more details and complexities of the natural world are incorporated into these models, their resolution and analytical solvability become increasingly difficult. This phenomenon is commonly called the "spherical cow of physics." It represents the trade-off between model simplicity and capturing the intricacies of reality. Researchers can make the systems more manageable and derive analytical solutions by assuming simplified shapes or disregarding certain complexities. However, it is crucial to recognize that such models may only partially capture the richness of real-world phenomena. Despite their limitations, physical models are valuable tools for gaining insights and understanding complex systems within the constraints of available resources and computational capabilities. They provide a foundation for further refinement, experimental validation, and the development of more comprehensive models in the pursuit of scientific knowledge.

A numerical method known as the finite elements method (FEM) approximates the behavior within each part of a system to tackle complicated engineering issues. In EDS systems, the FEM is frequently used to thoroughly examine electromagnetic fields, forces, and material characteristics. While advantageous in its ability to create detailed and accurate models, it is burdened by the drawback of increased computational time and resource requirements. This disadvantage stems from the necessity to compute the physical equations within each small mesh domain, which demands significant computational effort. However, the method excels in its capacity to generate a model that closely approximates the behavior of an original system, allowing for more comprehensive understanding of the complex dynamics of the system. Despite its computational challenges, the FEM remains valuable in conducting in-depth analyses and simulations of various physical phenomena.

The EDS system is effectively represented by applying DCT, which treats it as an electrical circuit comprising lumped parameters. This modeling approach considers various circuit characteristics that vary over time and space, governed by differential equations. As a result, it enables the dynamic and transient analysis of the system, ensuring precise force calculations and accommodating considerations of edge and end effects. Moreover, this methodology is rooted in the concept that the EDS system can be effectively simulated using an electrical circuit simulator. This topic is further elaborated upon in subsequent sections of this study.

Numerical methods are used to determine the behavior of several systems under various operating conditions. In this study, these methods are employed to solve mathematical equations iteratively. Specifically, the numerical method calculates the derivatives of mutual inductances between the onboard and guide coils. For this purpose, the centered derivative method is employed, which relies on determining the slope between two neighboring points around the desired evaluation point. It is important to note that numerical calculations inherently introduce some errors. Thus, meticulous care must be taken to select the most suitable method for the specific problem, with the aim of leveraging the significant advantage of fast processing speed offered by numerical calculations.

The Fourier transform analysis approach is utilized to decompose complex waveforms into their constituent sinusoidal components, enabling a thorough exploration of the spatial and harmonic properties of magnetic force and field distributions. This technique operates in the frequency domain, representing a time function of a signal as a combination of different frequencies. In the domain of EDS systems, the application of

this analysis is particularly notable in developing a comprehensive theory of moments for EDS magnetic levitation systems. This theory, incorporating double Fourier series and dynamic circuit principles, facilitates the calculation of angular derivatives of moments and forces, as extensively discussed by Knowles (1982) [26]. Fourier analysis achieves a deeper understanding and analysis of EDS systems, providing valuable insights into their behaviors and performances.

Experimental testing and measurements play a crucial role in studying the performance of EDS systems and validating theoretical models. These experiments use physical setups, apparatus, and data-gathering techniques to record and analyze forces, vibrations, and other relevant factors. While measuring devices inherently introduce some level of error and uncontrollable random variables exist in natural systems, it is through experimentation that theories are validated. In this work's case study, the experimental data obtained from the test track substantiated the feasibility of the proposed method. Integrating experimental data ensured more comprehensive understanding of, and validation of, the proposed approach, and strengthened its credibility and applicability in real-world scenarios.

Comparative Analysis of Related Studies

The intuitive model for eddy currents in thin sheets was presented by Maxwell [27] in 1972, predating the understanding of electron properties and serving as a foundational work in EDS. Smythe [28] described the fields within metallic plates using diffusion equations. However, these analyses were conducted solely for point magnetic charges. Paul [29] utilized the method of image currents to describe magnetic potential in perfect conductors.

Sinha [18] applied the image method and Newton's formula to calculate the resultant force on a coil moving over an infinite conducting plane. Slemon [30] derived the image force formula acting on a real coil due to its image, employing the conservation of energy theorem. Subsequent studies began to employ empirically obtained constants.

Reitz [31] established the relationship between lift and drag forces for various magnetic sources. Hill [32] demonstrated the need for different calculations of maximum drag force, based on the thinness/thickness, or penetration depth, of a metallic plate. Guderjahn [33] proposed a similar correction to Maxwell's recoil velocity.

The FEM is a numerical tool for solving Maxwell's equations, considering electromagnetic fields with specified boundary conditions and design geometries [34,35]. However, the method becomes computationally intensive when dealing with systems involving relative movements and temporal and spatial dependencies, due to the dynamic meshing required at each simulation step, particularly in 3D analyses. As shown by Gong et al. [21], efforts to circumvent the need for meshing at each step aim to streamline the computational process. Utilizing the vector magnetic potential boundary condition, they avoided the moving mesh of onboard superconducting (SC) coils. However, as the number of coils in a model increases, so does the computation time required to calculate the forces.

The Argonne Laboratory introduced the DCT for EDS systems in the 1990s. Considering the many components involved, its elegance and efficiency stem from representing an EDS system as a set of differential equations in matrix form. The focus is on directly calculating forces without additional transformations [36]. The primary challenge lies in determining the mutual inductances between any two coils, which depends on their relative positions, which continuously change due to train displacement. All physical quantities are derived from changes in magnetic energies stored in the system, dictated by the mutual inductances between onboard and offboard coils. Therefore, prior calculation of mutual inductance tables, obtained through numerical calculations, as proposed in this work, offers a means to significantly expedite the analysis of EDS systems.

Harmonic analysis employing Fourier transforms, coupled with numerical methods, is used to ascertain lift and drag forces on continuous sheet guideways, as demonstrated by Reitz and Davis [19] when investigating the motion of a rectangular coil on a conducting sheet. This technique is typically limited to two-dimensional steady-state analyses,

assuming infinite guideway conductor width. It may also apply to guideways with discrete coils [37].

Fujimoto et al. [38] provided the most cited experimental data on induced currents and generated forces in the EDS system of the Yamanashi test line. The values obtained from this paper are compared with those derived from the method presented herein. Experimental analysis remains a significant challenge in studying EDS systems, due to the high cost of SC coils and their requisite cooling systems. Consequently, most experiments utilize permanent magnets or Halbach arrangements as substitutes for SC coils. Additionally, linear test tracks are often substituted with rotational test benches to replicate high speeds, as observed in the study by Song et al. [39].

The main contribution of this work lies in developing a novel approach to calculating forces in null-flux EDS systems, offering a more time-efficient solution. This approach streamlines the investigation of EDS phenomena by simplifying the research process and eliminating the need for complex algorithms. Furthermore, it provides a straightforward method to study non-cross-connected null-flux systems, enabling rapid computation of induced currents and generation of forces using an electrical circuit simulator. By introducing this innovative approach, this work contributes to advancing the field of EDS and offers valuable insights into the behavior and performance of EDS systems.

### 2.2. The Basic Principles of an EDS System

An EDS system comprises three types of coils, as illustrated in part a of Figure 1. The SC coils are onboard, and the guideway coils include the levitation and propulsion coils. The levitation coils are figure-eight-shaped, known as 8-shaped or ground coils [36]. In the 1960s, physicists James R. Powell Jr. and Gordon T. Danby developed suspension systems based on electromagnetic forces [40]. They patented a null-flux suspension system to minimize drag forces, requiring less propulsion power. The most common configuration is displayed in parts b and c of Figure 1.

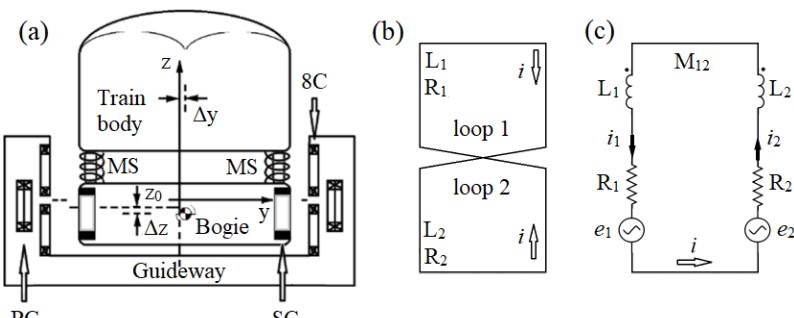

**Figure 1.** (**a**) Diagram of the main components of the JR–Maglev, Yamanashi test track. PC: propulsion coil. MS: mechanical suspension. SC: superconducting coil. 8C: figure-eight-shaped coil. (**b**) Sketch of the null-flux coils. (**c**) Equivalent electrical circuit. The numbers 1 and 2 refer to the upper and lower loops of the coil, respectively [41].

The 8-shaped coil is formed by connecting two coils in series, both of which are fed by the same magnetic field source, which is electrically isolated from any other element in the system. The operating principle is that the counter electromotive force generated by the upper loop (1) opposes that produced by the lower loop (2); hence, they are connected in reverse bias. The advantage is the absence of electrical current and power loss when the magnetic field source is centered relative to the coils. The 8-shaped coils are mounted vertically on both sides of a U-shaped track. This topology is known for its high lift–drag ratio, indicating high propulsion efficiency in the train [13]. An electric current is induced in the ground coil when a magnetic source passes below its center, generating the levitation force to balance the train's weight.

The case study focuses on the JR–Maglev, developed by the Central Japan Railway Company. JR Central plans to deploy the L0 Series train on the Chuo Shinkansen rail line

between Tokyo and Osaka, which is currently under construction. This study refers to data from the MLX01 series train measured on the Yamanashi test line due to difficulty in obtaining data from the current model, as provided in Table 1 [24,25,41]. The conventional notation used is depicted in part a of Figure 2. The Maglev's longitudinal movement aligns with the *x*-axis, representing the drag force, $f_x$. The transverse movement, or lateral displacement, follows the *y*-axis direction, representing the guidance force, $f_y$. Lastly, the vertical motion is associated with the *z*-axis, with $f_z$ representing the levitation force. Position $z_0$ is the equilibrium position, where the lift force equals the Maglev's weight force in magnitude, allowing it to move without touching the ground. The $\Delta y$ and $\Delta z$ represent the fluctuation during the train's movement in the *y* and *z* directions, respectively. The dimensions of the current SC and 8-shaped coils of the Yamanashi test track are shown in part b of Figure 2.

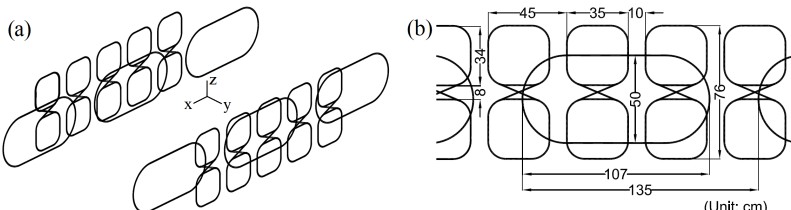

**Figure 2.** (**a**) Conventional notation used in Maglev systems. Sketch of SC and 8-shaped coils. (**b**) Dimensional drawing of the current 8-shaped and SC coils from the Yamanashi test track.

**Table 1.** Yamanashi test track parameters [24,25,41].

| Physical Parameter | Value | Unit |
|:---|:---:|:---:|
| Width × height, 8-shaped coil | 0.35 × 0.34 | m × m |
| Number of turns, 8-shaped coil | 24 | units |
| Resistance, 8-shaped coil (1 loop) | 8.99 [1] | mΩ |
| Inductance, 8-shaped coil (1 loop) | 287.26 [1] | μH |
| Mutual inductances [2] | 21.22 [1] | μH |
| Distance between centers [2] | 0.42 | m |
| Pole pitch, 8-shaped coil | 0.45 | m |
| Air gap between SC and 8-shaped coil | 0.185 | m |
| $z_0$ equilibrium position | 0.0366 | m |
| Width × height, SC coil | 0.963 × 0.5 | m × m |
| Magnetomotive force, SC coil | 700 | kAturn |
| Inductance, SC coil | 2.7 | H |
| Pole pitch, SC coil | 1.35 | m |

[1] These values were calculated analytically [41] and compared with those presented in the RTRI papers [24,25].
[2] Refers to the upper and lower loops of the 8-shaped coil.

### 2.3. The Mathematical Model of an EDS System

The DCT approaches an EDS system by considering space and time-dependent circuit parameters governed by differential equations. These equations are solved for time-domain currents, making the method suitable for dynamic and transient analysis through computer simulation. The electrical circuit in Figure 1 represents the simplest equivalent circuit for an 8-shaped coil interacting with a magnetic field source, such as that of the SC coils used in the JR–Maglev. The circuit features two branches in a mesh, each representing one of the loops with a resistance *R* and an inductance *L*. $M_{12}$ denotes the mutual inductance between the lower and upper loops of the 8-shaped coil. As it is a series circuit, the current in both loops is identical but flows in opposite directions, represented by *i*. The voltage sources, $e_1$ and $e_2$, signify the couplings between the SC coils onboard the train and the

null-flux coil loops on the track, expressed as current, displacement, and speed functions. The voltage equation for the equivalent electrical circuit is Equation (1) [36]:

$$e_1 - e_2 = 2Ri + 2(L - M_{12})\frac{di}{dt},\qquad(1)$$

shows the space–time dependence of the voltages and current induced in the null-flux coil.

The focus is on directly calculating forces. Physical quantities, such as energy, power, and forces, are expressed in their equivalent circuit parameters and spatial and temporal functions. The electromagnetic forces generated by the SC and the 8-shaped coils comprise three components. The drag force resists the train's movement, necessitating a propulsion force from the motor to overcome this force and enable displacement. The guidance force aligns the train on the track and prevents it from touching the guideway's sides. The lift force prevents the train from touching the ground and allows it to run without friction. Based on the conventional notation used in Maglev systems and illustrated in Figure 2, the drag force $\overrightarrow{f_x}$, the guidance force $\overrightarrow{f_y}$, and the lift force $\overrightarrow{f_z}$, are given by the Equation (2) [36]:

$$\overrightarrow{f_j} = I\,i\left(\frac{\partial M_{S2}}{\partial j} - \frac{\partial M_{S1}}{\partial j}\right)\hat{j},\qquad(2)$$

where $j = x, y, z$; $I$ represents the magnetomotive force of the SC coil, $i$ denotes the null-flux coil current, and $M_{S1}$ and $M_{S2}$ are the mutual inductances between one of the ground coil loops and the SC coil. Due to these couplings, the analytical resolution of this problem relies on numerical calculation.

### 2.4. The Computational Algorithm

The flowchart depicted in Figure 3 provides a comprehensive illustration of the sequential steps involved in this work, which are elaborated upon in subsequent sections. The primary objective is to present an overview of the implemented computational algorithm and the specific order of software utilization.

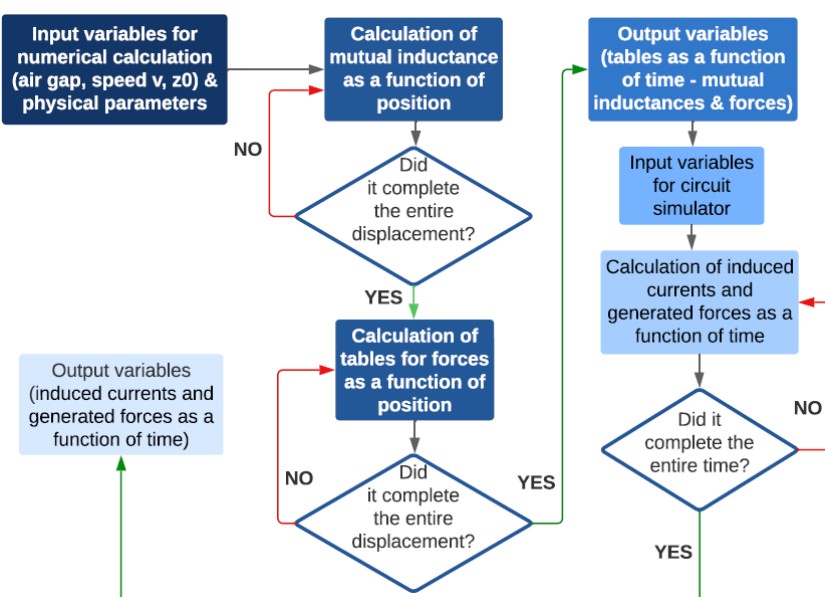

**Figure 3.** Calculation flowchart of this simulation model.

The input for this algorithm encompasses the physical parameters of the coils, specifically those outlined in Table 1, along with variables such as train speed, lateral air gap, and the train's relative "height" concerning the guideway coils. Only the self and mutual

inductances of the loops of the same null-flux coil and its resistance were calculated, utilizing Newmann's formula and Ohm's second law, respectively [41].

The software responsible for numerically calculating mutual inductances between onboard and guideway coils, including the *x*, *y*, and *z* derivatives of these inductances, receives the above inputs and generates output tables. These tables are a basis for computing the electrical circuit simulator's drag, levitation, and orientation forces.

The electrical circuit simulator, in turn, utilizes these pre-calculated tables as input data. By considering the mutual inductances as a function of time, the simulator models the magnetic coupling between the coils, effectively emulating real-world scenarios on a test track. As a consequence of this interaction, an induced current arises in the 8-shaped coil.

The interplay between the induced current and the original current from the SC coil, which serves as the source of the magnetic field, generates magnetic forces along the three Cartesian axes. The outputs of the electrical circuit simulator are the induced current, and these forces, which can be presented in tabular or graphical formats, catering to the user's preferences.

As a result of its numerical iterative nature, this computational algorithm substantially reduces the computational effort required, compared to conventional finite element simulations. This characteristic is visually demonstrated in Figure 4, which presents the hardware specifications employed in this study. Since it was a simulation work, a conventional desktop computer, and a laptop were utilized for the computational simulations. It is important to note that no specific hardware was developed in this research, as the focus was on the computational modeling and analysis of the EDS system. The significance of this work lies in its ability to streamline the research process for those investigating electrodynamic levitation phenomena.

**(a) Device specifications**

| | |
|---|---|
| Device name | DESKTOP-0KF4M6D |
| Processor | Intel(R) Core(TM) i5-8400 CPU @ 2.80GHz 2.81 GHz |
| Installed RAM | 8.00 GB (7.84 GB usable) |

**(b) Device specifications**

| | |
|---|---|
| Device name | DESKTOP-BO3UI48 |
| Processor | Intel(R) Celeron(R) CPU N2840 @ 2.16GHz 2.16 GHz |
| Installed RAM | 4.00 GB (3.89 GB usable) |

**Figure 4.** (**a**) Specifications of the hardware used for numerical calculation, a desktop computer. (**b**) Specifications of the hardware used for the simulation of the EDS circuit, a laptop.

*2.5. The Calculation of Mutual Inductance between SC and 8-Shaped Coils*

Equation (2) demonstrates that the forces developed in a null-flux EDS system result from the magnetic coupling between the magnetic field source and the null-flux coils. As the Maglev train moves, an interaction occurs between the onboard and outboard coils, expressed by variation in mutual inductance between the SC coil and the lower and upper loops of the 8-shaped coil. Traditional methods for calculating mutual inductance include the Newman formula, empirical formulae, and finite-element calculations [21,29,42]. In this work, the magnetic flux values, as functions of position, are determined by calculating the magnetic field generated by the SC coil through the Biot–Savart law, integrated over the area formed by the upper and lower loops of the 8-shaped coil [43]. For mathematical simplification, the SC coil was approximated to a rectangular shape with the same internal area as the actual coil, which had a racetrack shape, as shown in Figure 2. The flux of each side was determined separately, and the four fluxes were summed to obtain the total magnetic flux through the rectangular loop [44].

The simulations focused on a type of EDS system without cross-connections, using one 8-shaped coil and one SC coil. Wolfram®'s Mathematica software, version 12.1, performed the numerical calculations. The magnetic flux was calculated for a total distance of 32 m traveled by train, with 1-centimeter steps, totaling 3201 points, in the *x*-direction, from position −16 to +16 m. In the *y*-direction, the distance between the SC and 8-shaped coil centers was 0.185 meters, corresponding to the air gap shown on the Yamanashi test line. In the *z*-axis, the center of the SC coil was offset by 0.0366 meters below the center of the 8-shaped coil. This offset was the $z_0$ position. The mutual inductance as a function of position was then calculated using Equation (3) [44]:

$$M_{s,j}\left[x + \frac{\Delta x}{x}\right] = n * \frac{\phi_j[x + \Delta x] - \phi_j[x]}{I},$$ (3)

where $M_{s,j}$ represents the mutual inductance between the SC coil and the upper loop, $j = 1$, or lower loop, $j = 2$, of the 8-shaped coil; $\phi_j$ denotes the magnetic flux passing through the upper loop, $j = 1$, or lower loop, $j = 2$, of the 8-shaped coil; $n$ is the number of turns of one of the ground coil loops, and $I$ is the magnetomotive force of the SC coil, generating 3200 points. The graph of the mutual inductance as a function of distance can be seen in Figure 5.

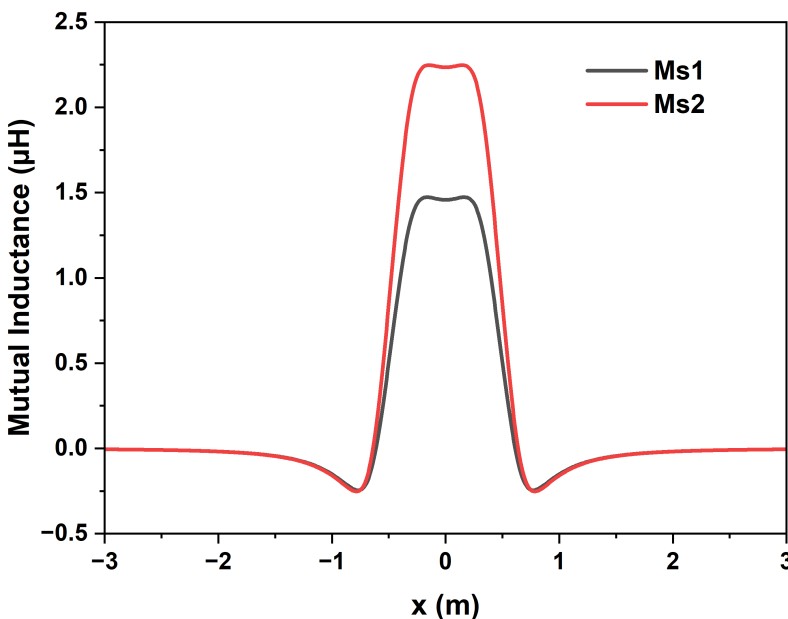

**Figure 5.** The mutual inductance between the SC and 8-shaped coils depends on their relative positions.

### 2.6. The Implementation of the Analysis Using a Circuit Simulator

The EDS system was simulated using Ansys® Twin Builder software, version 2022 R1. The circuit simulator's solver was set in transient mode, chosen due to the voltage source variation with the train's displacement, which depends on the train's speed. This resulted in a non-sinusoidal, time-varying source. Figure 6 displays the circuit implementation with one SC and one 8-shaped coil, with the physical parameters used in the simulation presented in Table 1.

In the implemented electrical circuit, the SC coil, acting as the primary, was represented by an ideal current source of 700 kA, along with the minimum resistance required by the software to run the simulation. The induced voltage in the 8-shaped coil served as the secondary of the implemented electrical circuit. All inputs were given as a function of time for the simulation. The vehicle speed was considered constant to verify if the simulation could be conducted in this manner, and the results were compared with those reported in the literature when introducing magnetic coupling as a data table as a function of time. Five Maglev displacement speeds were studied: 10, 50, 200, 298, and 600 km/h. These values

were chosen to align with Cai et al. (2020) [45] and Gong et al. (2021) [21] for comparison purposes only. Time was easily calculated as a function of speed and position.

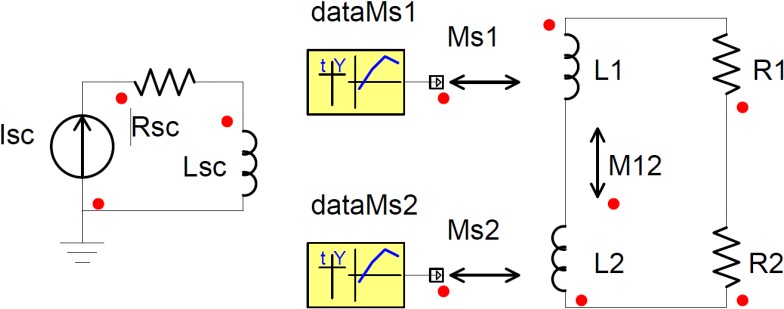

**Figure 6.** Circuit implemented in Ansys® Twin Builder software. The indices 1, 2, and sc refer to the upper and lower loops of the 8-shaped and the SC coil, respectively. Data: data tables with mutual inductances $M_{s1}$ and $M_{s2}$.

The value of magnetic coupling between the onboard and off-board coils, expressed by the mutual inductance, is low due to various factors, such as the distance between the coils and the fact that air served as the core of both coils, among other reasons. Precisely for this reason, the magnetic field powering the circuit must be intense, thus requiring the use of the SC coil. Another observation was that the magnetic coupling between the lower loop of the 8-shaped coil and the SC coil was greater than the coupling between the upper loop and the SC coil because the vertical deviation between the coils results in more magnetic flux in the lower loop than in the upper loop of the ground coil. Figure 7 presents the induced currents obtained at the five operational speeds, which agreed with the findings in the literature that the induced currents become positive with increasing speed [21,36,45].

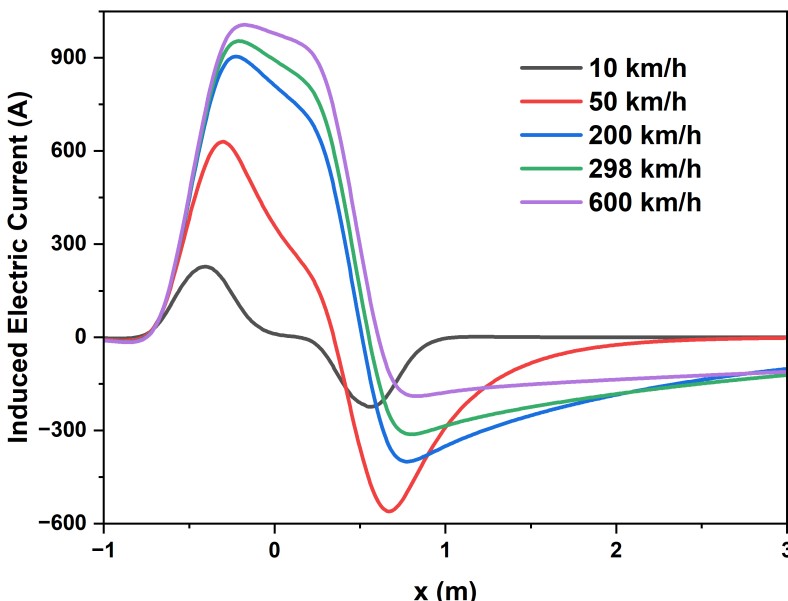

**Figure 7.** Electric current induced in the 8-shaped coil fed by one SC coil for five different speeds.

### 2.7. The Electromagnetic Forces

The displacement of a single SC coil in front of an 8-shaped coil was simulated to calculate the electromagnetic forces. The drag, guidance, and levitation forces, according to Equation (2), depend on the derivatives of the mutual inductance concerning the *x*-, *y*-, and *z*-axis, respectively. These data were also calculated numerically. The centered derivative method was used to calculate the derivative of mutual inductances. For the drag force, the mutual inductance was calculated twice for all 3201 points along the 32 m, offset

by $-1$ and $+1$ mm, $\Delta x$, concerning the longitudinal axis. The drag force was determined by Equation (4):

$$\overrightarrow{f_x} = I\,i\left(\frac{M_{s,1}[x+\Delta x]-M_{s,1}[x-\Delta x]}{2\Delta x}-\frac{M_{s,2}[x+\Delta x]-M_{s,2}[x-\Delta x]}{2\Delta x}\right)\hat{\imath} \tag{4}$$

and the same procedure was performed for the guidance and lift forces, with their equations being Equations (5) and (6):

$$\overrightarrow{f_y} = I\,i\left(\frac{M_{s,2}[y+\Delta y]-M_{s,2}[y-\Delta y]}{2\Delta y}-\frac{M_{s,1}[y+\Delta y]-M_{s,1}[y-\Delta y]}{2\Delta y}\right)\hat{\jmath} \tag{5}$$

$$\overrightarrow{f_z} = I\,i\left(\frac{M_{s,2}[z+\Delta z]-M_{s,2}[z-\Delta z]}{2\Delta z}-\frac{M_{s,1}[z+\Delta z]-M_{s,1}[z-\Delta z]}{2\Delta z}\right)\hat{k} \tag{6}$$

where the differences between (4), (5) and (6) were due to the positioning of the coils relative to the coordinate system. The imported tables for the circuit simulator were generated similarly to those for the mutual inductance between the upper and lower loops of the null-flux coil and the SC coil as a function of time for each speed, using Wolfram®'s Mathematica software.

Figure 8 displays the output to Ansys® Twin Builder user for the calculated forces at a movement speed of 600 km/h, showcasing the circuit with the plot of these forces. The results were consistent with those presented in various classical works in the literature, demonstrating that the method is suitable for the study of EDS systems [21,36,45]. Figure 9 shows the drag force, Figure 10 the guidance force, and Figure 11 the lift force generated in this system under different operational speeds of the Maglev vehicle.

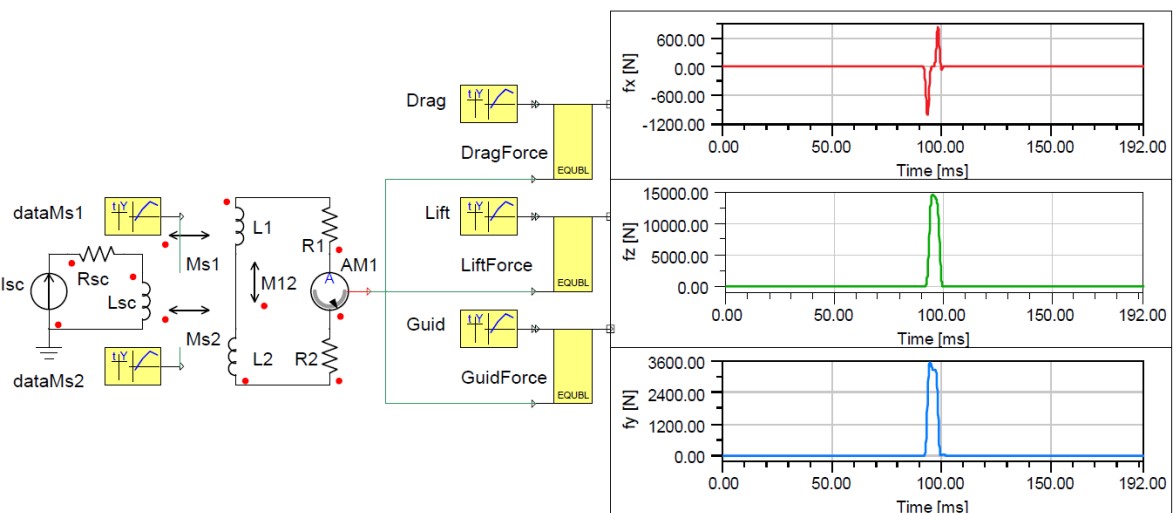

**Figure 8.** The electrical circuit implemented in the Ansys® Twin Builder software, along with the respective force curves generated as a function of time for a cruising speed of $v = 600$ km/h. The red, green, and blue lines represent drag, lift, and guidance forces, respectively.

Figure 7 demonstrates that the induced currents became positive with increasing speed. As the leading edge of the SC coil approached the 8-shaped coil, the induced current increased. Conversely, it decayed and turned negative as the trailing edge of the SC coil moved away from the 8-shaped coil. This alternation of the current signal induced by the same SC coil implied that it received a repulsive impulse when approaching and an attractive impulse when moving away. Consequently, the lift force oscillated between positive and negative values at one period. As the speed increased, the effect of the oscillating lift force ceased, as seen in Figure 11. The current induced in the 8-shaped coil no longer changed sign, and the lift force remained consistently positive.

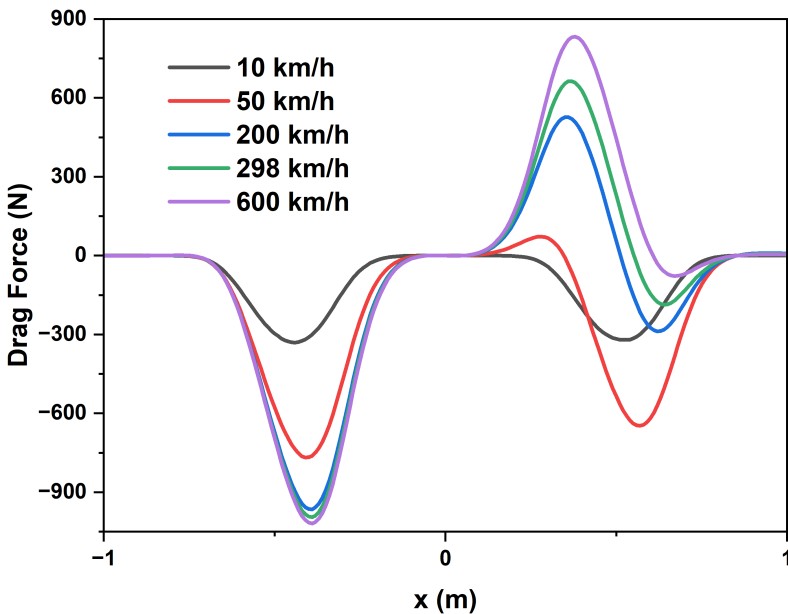

**Figure 9.** Drag force for five different speeds.

The change in the behavior of the induced electric current was due to the longitudinal force component $f_x$, which consists of a conservative part and a dissipative part [13]. The dissipative term, which represents the magnetic drag of the system over a period, can be observed in Figure 9. It always decelerates the train, directly proportional to the electric currents in both coils, and inversely proportional to the train's speed. As a characteristic of EDS systems, the drag force reaches its maximum value at low speeds and tends to have a constant value at high speeds [32,46].

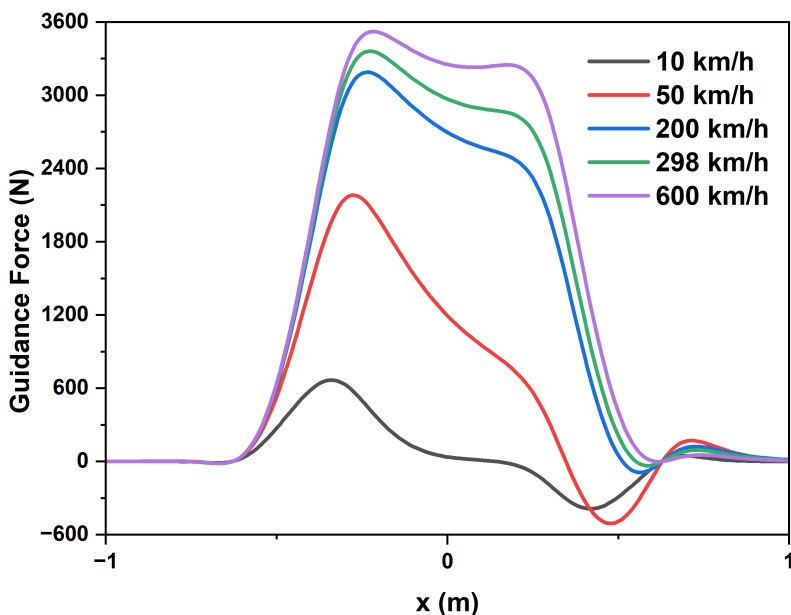

**Figure 10.** Guidance force for five different speeds.

Examining the central part of the last curve in Figure 8 reveals that the guidance force, depicted in Figure 10, had a higher value, which tended to decrease. It is important to note that only one side of the system was plotted. A null-flux coil on the opposite side of the track, with the same 0.185-meter gap, would generate an equal guidance force in the opposite direction, preventing the vehicle from moving sideways.

The SC coils, designed in a racetrack format, are intended to withstand the electromagnetic forces generated in the EDS system. The lift force acts on the straight horizontal lines of the SC coils, while the propulsion and drag forces act on the curved sides [36].

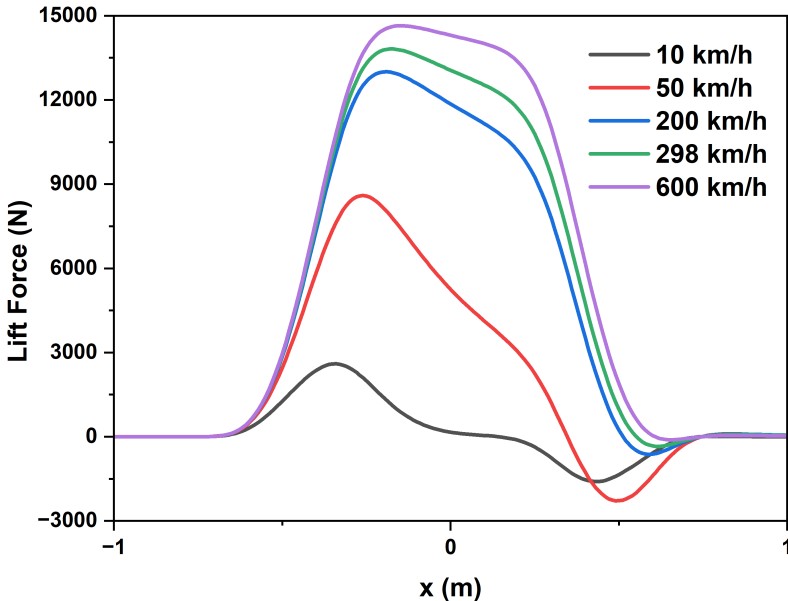

**Figure 11.** Lift force for five different speeds.

We examined the effect of varying another circuit parameter, that being the distance between the onboard and offboard coils. The lateral air gap, typically characterized by a value of 0.185 m, was adjusted within the range of 0.170 m (approximation) to 0.200 m (greater distance between the coils). The speed was kept constant at 600 km/h.

Figure 12 illustrates the impact on the induced current in the 8-shaped coils resulting from this variation. It can be observed that as the distance between the coils decreased, the induced current increased, when at the same speed. Figure 13 depicts the effect of this variation on the drag force, revealing a signifcant increase. Similarly, Figure 14 demonstrates the direct influence on the levitation force. However, in a railway bogie, one side pulls away from the guideway while the other approaches. The average magnitude of the drag and levitation forces, with only the lateral displacement varying, closely resembles the value of these forces when the Maglev moves without lateral displacement in a non-cross-connected system. Nevertheless, as these forces are vectorial and are applied at different points on the railway bogie, a rotational effect occurs on the train.

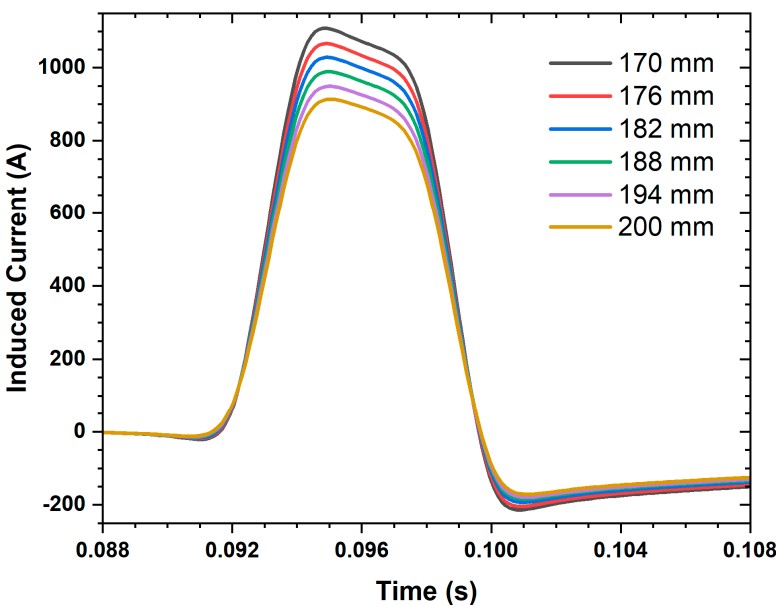

**Figure 12.** The electric current induced in one 8-shaped coil fed by one SC coil for six different air gaps.

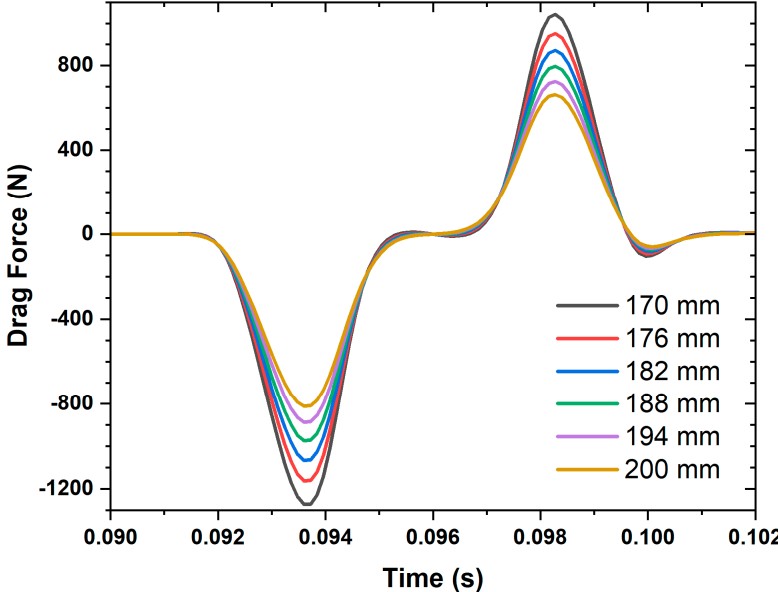

**Figure 13.** Drag force in one 8-shaped coil fed by one SC coil for six different air gaps.

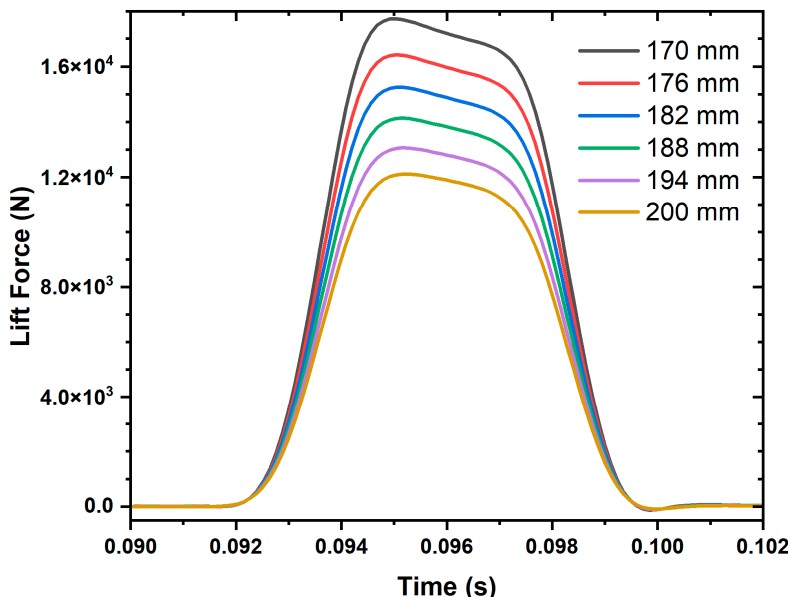

**Figure 14.** Lift force in one 8-shaped coil fed by one SC coil for six different air gaps.

Figure 15 provides valuable insights into the influence of varying the lateral air gap on the guiding force, highlighting its high significance. The closer the coils, the greater the guidance force trying to push them apart. In the EDS system, where the ground coil is distributed on both sides of the U-shaped track, the total guidance force is determined by the difference in the guidance force generated by the 8-shaped coil on each side of the bogie. When one side of the bogie is positioned close to the guideway, the other is at an equivalently greater distance.

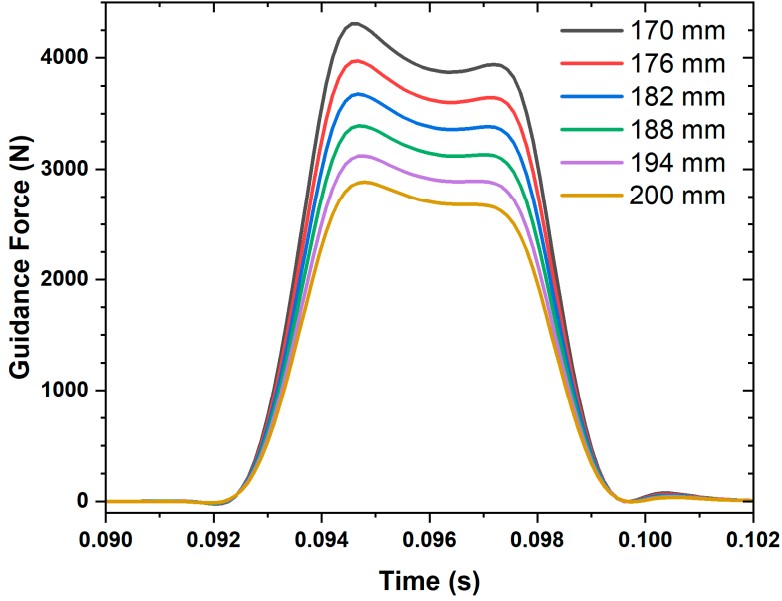

**Figure 15.** Guidance force in one 8-shaped coil fed by one SC coil for six different air gaps.

### 2.8. The Model Validation

Calculations were performed to validate the presented model's accuracy and to determine the electromagnetic forces generated by the interaction of 4 SC coils on the bogie and 20 null-flux coils on the guideway. The analysis focused on the average electromagnetic forces as a speed function, ranging from $v = 0$ km/h to 600 km/h. The obtained results were

compared with the data measured from the Yamanashi test line [38]. Figure 16 displays the values of drag and levitation forces, while Figure 17 illustrates the guidance force.

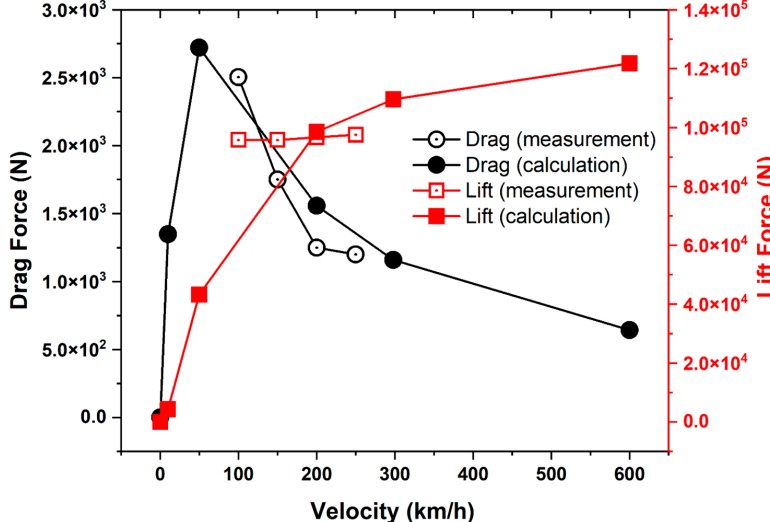

**Figure 16.** Levitation and drag forces as a function of displacement speed. A comparison between the model utilizing the circuit simulator and experimental measurements conducted on the Yamanashi test line. The configuration consists of four SC coils onboard and twenty 8-shaped coils on the guideway.

In Figure 16, the characteristic curves of EDS system can be observed through the levitation and drag forces. The levitation force exhibited an increasing trend with speed until it reached a maximum value. This behavior is attributed to the simultaneous increase in induced voltage in the ground coil and the coil's effective resistance, due to the metallic conductor's skin effect. Consequently, the induced current in the coil increased at a slower rate or stopped increasing, resulting in saturation of the suspension force at higher speeds. The experimental data indicated that this saturation speed occurred at approximately 100 km/h. The critical speed, which depends on the magnetic field strength in the air gap and the vehicle weight, refers to the speed at which the vertical force of the vehicle becomes positive [2]. It is also called lift-off speed. Once the system achieves levitation, it remains sustained as long as the onboard SC coils operate stably.

The drag force exhibited a sharp increase up to approximately 50 km/h, followed by a gradual decrease after reaching the peak's speed of the electromagnetic drag force. This phenomenon indicates that less energy is lost as the train's speed increases. The reduction in energy loss can be attributed to the fact that if the time it takes for a SC coil to cross the null-flux coil is shorter than the current decay time of the ground coil, the null-flux coil does not have sufficient time to dissipate the stored energy within the Maglev system. Thus, the variation of the drag force with velocity, as shown in Figure 16, can be explained.

Figure 17 portrays the relationship between the guidance force and speed. As speed increased, the guidance force gradually rose. However, after reaching a speed of 200 km/h, the guidance force increased at a slower rate. This behavior mirrored the relationship observed in the levitation force–speed characteristics. At higher speeds, the induced current remained relatively stable due to the skin effect, leading to a tendency for the guidance force to approach saturation.

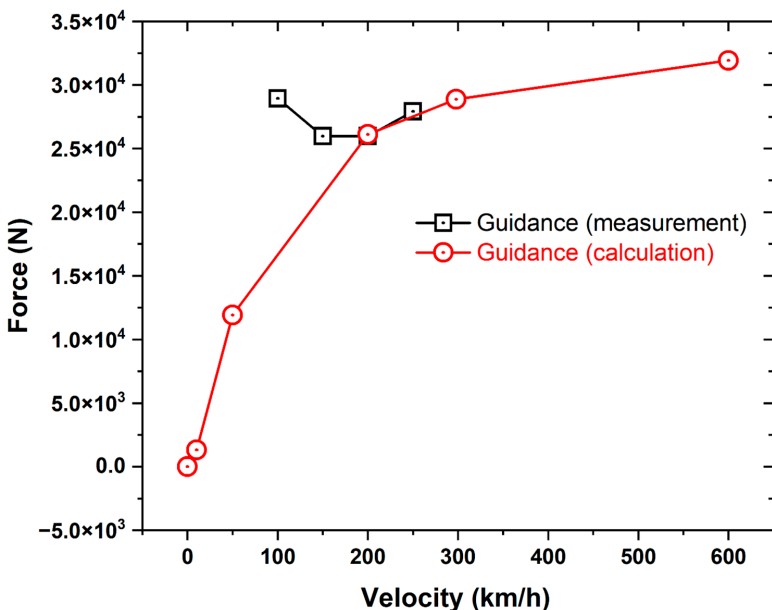

**Figure 17.** Guiding force as a function of train travel speed. A comparison between the model using the circuit simulator and experimental measurements obtained from the Yamanashi test line. The setup includes four SC coils onboard and twenty 8-shaped coils on the guideway.

The simulation results, as depicted in Figures 16 and 17, exhibited a notable disparity between the calculated values and the measured data, which could be attributed to the approximations made in the numerical models regarding the coil parameters. Specifically, the SC coil, originally with a racetrack shape, was approximated as a rectangle with an equivalent internal area. Similarly, the 8-shaped coil was approximated using two rectangular loops, disregarding the curved ends of the actual coil. Moreover, both coils were considered one-dimensional, that is, without thickness. Additionally, the proposed model did not consider certain aspects, such as the shield of the SC coil. As a result, discrepancies arose in the mutual inductances between the onboard and off-board coils and the induced current in the 8-shaped coil. It is important to note that the electromagnetic forces in the analytical model were directly proportional to the induced current and the derivative of the mutual inductance concerning displacement. Nevertheless, despite these discrepancies, the numerical results and measurement data exhibited significant agreement regarding primary characteristics, thereby validating the numerical model.

### 2.9. Three Superconducting Coils

The approach can be extended to include three SC coils, as illustrated in Figure 18. This spatial arrangement was selected due to its resemblance to the magnetic field generated by Halbach array permanent magnets [12].

The polarities between two adjacent SC coils were reversed, similar to the JR–Maglev configuration. This reversal helps to reduce the magnetic field inside the train by creating a magnetic path between adjacent coils. Consequently, the magnetic field levels within the car remain within safe limits for passengers, and less robust magnetic shielding is required [13].

The magnetic coupling tables for the two new coils were shifted by a polar pitch of 1.35 m, coil $p1$ leading for coil $m1$ lagging. The same procedure was applied in calculating the drag, guidance, and lift forces presented in the table. Ultimately, the table imported for the computing of each of the three forces was the superposition of the three tables, one data table for each of the three SC coils. The results, at a speed of 600 km/h, can be observed in Figure 18, which supports the analysis of more complex null-flux EDS systems.

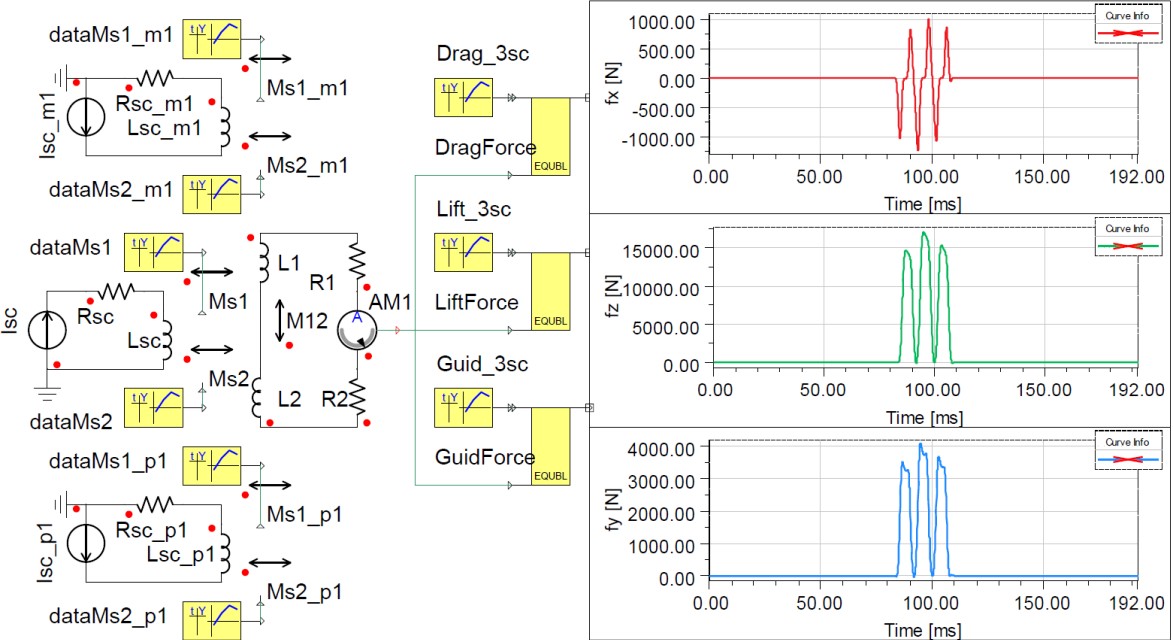

**Figure 18.** The electrical equivalent circuit of an 8-shaped coil powered by three SC coils, forming a null-flux EDS system at a cruising speed of $v$ = 600 km/h, was implemented in the Ansys® Twin Builder software, which also displays the respective force curves generated as a function of time. The red, green, and blue lines represent drag, lift, and guidance forces, respectively.

Figure 18 was plotted to demonstrate the potential for increased system complexity, either on both sides of the guideway or with additional 8-shaped coils. The mutual inductances between each newly added coil and the previous ones had to be calculated, always utilizing the symmetry and periodicity properties of the system by shifting the calculated values by one polar pitch. It was even possible to cross-connect both sides of the guideway, making the electrical circuit equivalent to a null-flux levitation and guidance system.

## 3. Discussion

This study presents the analysis and advantages of a null-flux coil system for Maglev suspension, highlighting its potential benefits and limitations compared to previous studies and working hypotheses. One significant advantage of the null-flux coil system is its lack of dependence on a feedback control loop. Its configuration ensures it remains in a dynamic equilibrium position $z_0$. This feature simplifies the overall system while maintaining effective control. Furthermore, the null-flux coil system exhibits high vertical rigidity, resulting in minimal vertical displacement of the train when transitioning from on-load to full-load states. However, this system also possesses some drawbacks, such as low lateral stiffness. This limitation restricts the system's tolerance for lateral deviations, as large lateral displacements can generate torques in the vehicle, potentially compromising its stability. To address this issue, the suspension and guidance system, known as the cross-connected null-flux system, was developed to enhance stability during operation, and is already being analyzed for a study to follow using an electrical circuit simulator.

Figures 7–11 and 18 provide valuable insights into the EDS system's performance. Figure 7 confirms that induced currents become positive with increasing speed. Figure 8 shows that the method suits EDS system studies, and Figure 18 demonstrates the potential for increased system complexity. Figure 9 highlights the magnetic drag of the system, which is proportional to the electric currents in both coils and inversely proportional to the train's speed. The guidance force, which increases and decreases similarly to the levitation force, but with a much lower value, is illustrated in Figure 10. At the same time, Figure 11 depicts the lift force, which oscillates between positive and negative values at lower speeds but

remains consistently positive at higher speeds, due to changes in induced current behavior. Together, these figures provide a comprehensive analysis of the EDS system's performance and potential for improvement.

Figure 12 illustrates the effect of varying the lateral air gap on the induced current in the 8-shaped coils. As the distance between the coils decreases, the induced current increases. Figures 13 and 14 demonstrate that this variation has minimal impact on the drag force and negligible influence on the levitation force, respectively. Figure 15 depicts the influence of the lateral air gap on the guidance force. Figures 16 and 17 corroborate the feasibility of the proposed method. The analysis of these figures highlights the unique approach adopted in this study, which aims to simplify the research process in electrodynamic levitation.

The calculation of mutual inductance and its derivatives using the Biot–Savart equation provides a solid foundation for the numerical calculation of mutual inductance as a function of magnetic flux variation. By utilizing the circuit simulator, the complexity of solving electric circuit equations and determining forces is significantly reduced. This streamlined approach simplifies the analysis and allows the exploration of a cross-connected circuit, which will be the focus of a subsequent paper, without the need for complex harmonic approximations or other intricate considerations. Moreover, the flexibility of the circuit simulator enables electrical modifications and the introduction of additional elements, which align with a research topic of a further study. Together, these factors contribute to the innovative nature of the work and its potential to simplify and advance research in electrodynamic levitation.

One significant advantage of the analytical study is its time efficiency. The delay in the process lies in calculating mutual inductances and their derivatives to form force calculation tables. However, the simulation takes up to 10 s and can be run countless times after uploading the tables to the software. Another benefit is the ease of making changes with just one click. To switch from a non-cross-connected to a cross-connected system, the two null-flux coils are wired opposite each other on the guideway without needing to make any approximations to obtain induced currents in the 8-shaped coils, as reported by He et al. (1994)'s [36]. However, it should be noted that the force tables differ in this case.

The results obtained in this study support the advantages and limitations of the null-flux coil system, aligning with previous research and theoretical expectations. The findings also offer valuable insights into the system's performance and demonstrate electrical circuit simulators' effectiveness in analyzing EDS systems. The approach employed in this study facilitates more efficient system modifications and evaluations by streamlining the analysis process.

In light of these results, future research can focus on further improving the performance of the null-flux coil system by analyzing the cross-connected system. Additionally, researchers may explore novel applications for the null-flux coil system in other areas of transportation, as well as investigate the potential integration of this system with other emerging technologies. Ultimately, the findings of this study contribute to the ongoing development of Maglev systems, promoting more efficient, reliable, and sustainable transportation solutions.

## 4. Conclusions

This study presents an analytical model for analyzing a null-flux suspension Maglev system using a commercial electrical circuit simulator and DCT. This method allows for dynamic and transient analysis of EDS systems, producing results consistent with previous studies in the literature. One significant advantage of this approach is its time efficiency, as the simulation takes only 10 s to run and can be run countless times after uploading the force calculation tables to the software. This method also enables the efficient implementation of modifications and the rapid analysis of their effects, including introducing other elements. Furthermore, changes can be made with just one click, simplifying the system modification. The results of this study demonstrate the effectiveness of the analytical calcu-

lation approach, which offers a faster computation time than alternative methods, such as finite element analysis, while still producing accurate results. The reduced computation time required to run the simulation makes this approach particularly advantageous for the designing and optimizing of EDS systems. Experimental data from the Yamanashi test line were incorporated into the study. These empirical measurements validated and enhanced the analysis, providing real-world evidence to support the findings. By integrating qualitative data, the study gains credibility and offers a more comprehensive assessment of the Maglev system's null-flux suspension performance and behavior. Including experimental data strengthens the study's impact and contributes to the ongoing development and optimization of EDS systems.

**Author Contributions:** All authors contributed extensively to the work presented in this paper. Conceptualization, methodology, software, writing—original draft preparation, T.N.F.; writing—review and editing, H.L., H.Z., H.S., L.L. and Z.D.; supervision, L.L.; project administration, Z.D. All authors have read and agreed to the published version of the manuscript.

**Funding:** This work was supported by the Science and Technology Partnership Program, Ministry of Science and Technology of China under Grant KY201701001.

**Institutional Review Board Statement:** Not applicable.

**Informed Consent Statement:** Not applicable.

**Data Availability Statement:** By the data availability guidelines of the journal, it is noted that the files exemplifying the implementations conducted in this paper can be accessed at the following link: https://drive.google.com/drive/folders/1R6wFKVyrmOWp4IxKhVuX0Bh8HyYt2wr5?usp=sharing. This link provides access to sets of publicly archived data that were analyzed or generated during the study. Authors are encouraged to share their research data to promote transparency and facilitate further investigation.

**Acknowledgments:** The authors would like to express sincere gratitude and appreciation to Richard Magdalena Stephan for his invaluable insights and constructive feedback, which have greatly enhanced the quality and accuracy of this paper. His profound knowledge and expertise in magnetic levitation were instrumental in shaping the direction of this research and refining the arguments presented herein. The authors are genuinely indebted to Stephan for his time, dedication, and unwavering commitment to scholarly excellence.

**Conflicts of Interest:** The authors declare no conflict of interest. The funders had no role in the design of the study; in the collection, analyses, or interpretation of data; in the writing of the manuscript; or in the decision to publish the results.

## Abbreviations

The following abbreviations are used in this manuscript:

| | |
|---|---|
| 8-shaped coil, 8C | Figure-eight-shaped coil |
| DCT | Dynamic Circuit Theory |
| EDS | Electrodynamic Suspension |
| EMS | Electromagnetic Suspension |
| FEM | Finite elements method |
| HTS | High-temperature superconducting Suspension |
| JR Central | Central Japan Railway Company |
| JR-Maglev | Japan Railway Maglev system |
| L0 Serie | Levitation 0 Serie |
| Maglev | Magnetic levitation |
| MLX01 | Magnetic Levitation Experimental 01 |
| MS | Mechanical suspension |
| PC | Propulsion coil |
| RTRI | Railway Technical Research Institute |
| SC | Superconducting coil |

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
