# Peer review of "Electrical Circuits Simulator in Null-Flux Electrodynamic Suspension Analysis"

_applsci, doi:10.3390/app13116666_

Round 1

Reviewer 1 Report

This manuscript employed an electrical circuit simulator to investigate an electrodynamic suspension system for passenger rail transport applications. Major issues are associated with the manuscript which must be addressed before further processing of the manuscript. 

1. Abstract section doesn’t provide information about problem selected. Abstract can be written in more comprehensive and focused manner. Abstract, summarize the numerical results of proposed work, and discuss how it outperforms existing works.

2. Related work should be mentioned in more detailed way by highlighting the comparative analysis in tabular manner. What are the unique features of this study compared to the existing works? A ‘Research Gap’ section should incorporate which will states the purpose of the study.

3. Contributions should be highlighted in bullet points and justified.

4. Methodology section is not clear in present form. A flowchart should incorporate which represent the various steps of the proposed work.

5. A comparative analysis with the similar studies is missing which must be incorporated and compared to the novelty of the proposed work.

6. Authors must incorporate the picture of hardware setup. 

7. Conclusion also required presenting in more quantitative manner based on the obtained results.

8. Authors selected 5 different train speeds, what is the basis for the selection of these speeds. 

9. What are the assumptions which are considered in the proposed work, if practically implemented. 

Authors must thoroughly review the whole manuscript for improving the language of the manuscript.  

Author Response

Dear REVISER 1,

We have carefully considered all your comments and suggestions and made substantial revisions to address their concerns. This cover letter will summarize the major revisions made in response to your feedback.

  1. The abstract has been thoroughly revised to provide comprehensive information about the selected issue, summarize the results of the proposed work, and discuss how it surpasses or complements existing research.
  2. The paper now provides a more detailed discussion of related works, highlighting the unique features of this study compared to existing literature. These distinctive aspects have been introduced in the 'Research Gaps' section to state the study's purpose clearly. The primary objective of our work is to simplify the research process for those studying electrodynamic levitation. We achieve this by calculating the mutual inductance and its derivatives using the Biot-Savart equation. Subsequently, the mutual inductance is determined as a function of the magnetic flux variation within the internal area of the turns of the 8-coils through numerical calculations. Using a circuit simulator streamlines our work by solving electrical circuit equations and providing solutions for the forces. It also allows us the flexibility to analyze cross-connected circuits, which will be the focus of our future work, simply by connecting electrical wires without the need for complex harmonic approximations. Furthermore, the circuit simulator enables us to introduce additional elements into the circuit, which is a crucial aspect of one of our co-authors' thesis.
  3. We aim to emphasize and justify the point-to-point contributions made in this study.
  4. We have presented the methodology more clearly. To aid understanding, a flowchart has been incorporated to illustrate the various stages of the proposed work.
  5. In the introduction and the Research Gap section, we have included a comparative analysis with similar studies to highlight the originality and novelty of our work.
  6. We included a photograph.
  7. The conclusion has been rewritten to present the obtained results effectively.
  8. We have explicitly mentioned the values of the five speeds chosen for the simulations, which are the same as those used in the paper by researcher Guantong Ma. These values allow for a direct comparison with his results.
  9. The assumptions considered in our proposed work have been highlighted. Since it involves numerical calculations, the computational effort is significantly smaller than finite element simulations. The significance of this work lies in its ability to simplify the research process for those investigating electrodynamic levitation. By calculating the mutual inductance and its derivatives using the Biot-Savart equation, and subsequently determining the mutual inductance as a function of the magnetic flux variation within the inner area of the loops of the 8-coils through numerical calculations, we streamline the research process. The circuit simulator solves the electric circuit equations and provides solutions for the forces. It also enables the analysis of cross-connected circuits, which will be explored in a subsequent paper, simply by connecting electrical wires without the need for complex harmonic approximations. Additionally, it facilitates the electrical modification of the circuit by introducing other elements, which is the main focus of one of our co-authors' thesis.

We have carefully considered all the reviewers' comments and made the necessary revisions to address their suggestions. These revisions have significantly improved our manuscript's quality and scientific value. We hope the reviewers find our revised work satisfactory and allow us to contribute to the journal.

Thank you for your attention and consideration.

Sincerely

The Authors

Reviewer 2 Report

1. Each acronym should be introduced in details in the first presentation. The acronym of EDS in the Abstract is not introduced.

2. The main parts of a suitable introduction including the research motivation, the literature review, the necessity of the research based on challenges of the literature, the novelty and main contributions of the paper, have not been considered in this manuscript.

3. How the results provided by the proposed simulator can be verified? From my point of view, the case studies should be solved by some numerical models based on harmonic analysis via Fourier transform or finite element methods.

Author Response

Dear REVISER 1,
We have carefully considered all your comments and suggestions and made substantial revisions to address their concerns. This cover letter will summarize the major revisions made in response to your feedback.

  1. We have ensured that each acronym is clarified before its usage.
  2. The introduction has been extensively rewritten to align with a scientific text's requirements, incorporating the research's motivation, a comprehensive literature review, the need for further analysis based on existing challenges, the novelty of our work, and the main contributions of the article.
  3. We have introduced a method in the paper to quantify the results provided by our proposed simulator by comparing them with experimental values measured in the Yamanashi test line. To facilitate this verification process, we have included a figure displaying the values of drag and levitation forces and another figure showcasing the values of the guidance force. These comparative results provide a means to evaluate the accuracy of our simulator. In contrast, employing finite element methods would require a significantly longer time for simulations to be completed. We sincerely appreciate your efforts in evaluating our work thus far.

We have carefully considered all the reviewers' comments and made the necessary revisions to address their suggestions. These revisions have significantly improved our manuscript's quality and scientific value. We hope the reviewers find our revised work satisfactory and allow us to contribute to the journal.

Thank you for your attention and consideration.

Sincerely,

The Authors

Reviewer 3 Report

Dear authors,

This article presents an interesting topic. However, I have following suggestion for improvement.

1. The novelty in the introduction section is not properly explained/highlighted therefore significance of the work is unclear. In the current version, I can only see calculation of a parameter is presented as novelty, which is insufficient for a quality journal article.

2. In Table 1. most of the values are calculated based upon the methods already published. This further makes the contributions of this study unclear and insufficient.

3. The article claims that mutual inductance is calculated whereas its equations are taken from ref [27] and others. This nullifies the claim. Please explain.

4. The results do not show any comparisons with the previous studies utilizing similar techniques or other methods to show the improvements, if any. This will also help to demonstrate the effectiveness of this study.

5. Conclusion should report quantitative significance of the content presented in the article. 

Thanks

Fine

Author Response

Dear REVISER,

We have carefully considered all your comments and suggestions and made substantial revisions to address their concerns. This cover letter will summarize the major revisions made in response to your feedback.

  1. As per your suggestion, we have thoroughly rewritten the introduction to explain and highlight the novelty of our work. In the current version, we emphasize that the purpose of our study is to simplify the research process for those investigating electrodynamic levitation. Our approach involves calculating the mutual inductance and its derivatives using the Biot-Savart equation, followed by determining the mutual inductance as a function of the magnetic flux variation within the inner area of the loops of the 8-coils through numerical calculations. Using a circuit simulator simplifies the work by solving electric circuit equations and providing solutions for the forces. Additionally, it offers the freedom to analyze cross-connected circuits, which will be the subject of a subsequent paper, simply by connecting electrical wires without the need for intricate harmonic approximations. Moreover, it allows for the electrical modification of the circuit by introducing other elements, which aligns with the focus of one of our co-authors' thesis.
  2. Table 1 presents the physical parameters of the Yamanashi test track. We have included the self and mutual inductances, along with the resistance, which was calculated in the master's thesis of one of the authors using Newmann's and Ohm's second law, respectively. These values are provided to facilitate the reproducibility of calculations by other researchers.
  3. The calculation of the mutual inductance is based on equations derived from references [27] and others. The reference equation mentioned adapts the Biot-Savart law formulated for numerical methods. In our approach, we approximate the superconducting coil as a rectangle, and the formula calculates the magnetic field generated by each of the four sides of this rectangle. The magnetic field is then integrated as a function of the internal area of the null-flux coil loops to determine the mutual inductance between the superconducting coil and the loops of the 8-shaped coil.
  4. Our qualitative analysis compares the values obtained for the forces using our analytical method with the experimental data from the Yamanashi test track, presented in two figures in the paper. Conducting a comparative analysis using other ways, such as finite element analysis, would require a significantly longer time. Hence, we are grateful for your evaluation of our work thus far.
  5. The conclusion has been rewritten to effectively convey the significance of the content presented in the article.

We have carefully considered all the reviewers' comments and made the necessary revisions to address their suggestions. These revisions have significantly improved our manuscript's quality and scientific value. We hope the reviewers find our revised work satisfactory and allow us to contribute to the journal.

Thank you for your attention and consideration.

Sincerely,

The Authors

Reviewer 4 Report

1. What is the main question addressed by the research?

The authors present a research of an electrical circuit simulator to investigate an electrodynamic suspension system for passenger rail transport applications. They are focused into null-flux suspension system utilizing figure-eight-shaped coils, and their aim is to characterize the three primary electromagnetic forces generated in EDS. The presented paper study an analytical model for analyzing a null-flux suspension Maglev system using a commercial electrical circuit simulator and dynamic circuit theory.

2. Do you consider the topic original or relevant in the field? Does it address a specific gap in the field?

The topic is original, and I think that the research in this paper is relevant to the scientific orientation of the journal: Applied Sciences (ISSN 2076-3417); Section: Energy Science and Technology; special issue: Driving Automation Systems and Connectivity for a Sustainable Mobility. I have no doubts that the reviewed paper in related to the topic of analysis and simulation of electrodynamic suspension system for passenger rail transport applications.

3. What does it add to the subject area compared with other published material?

The authors present a very good review of the reference and previously published papers and patents. The authors based their research into two main previous research of the levitation coil – a developed suspension systems based on electromagnetic forces by physicists James R. Powell Jr. and Gordon T. Danby and Maglev system. Authors present a simulation model, based on conventional simulation software and this is the main advantage, that this model can be used and applied for investigation of such systems. The EDS system is simulated using Ansys Twin Builder software. The circuit simulators solver is set in transient mode, chosen due to the voltage source variation with the train’s displacement, which depends on the train’s speed.

4. What specific improvements should the authors consider regarding the methodology? What further controls should be considered?

       The presented paper is very well structured considered from scientific and practical point of view. The authors present a mathematical and simulation model of an electrodynamic suspension system for passenger rail transport applications. This is important area of research and I mean that it has a significant importance for the theory and practice. I suggest to authors to present or may be to describe very well practical benefits that they obtained from the carried research. I haven’t other recommendations and I mean that the paper doesn’t need any other changes in methodology or improvements.

       5. Are the conclusions consistent with the evidence and arguments presented and do they address the main question posed?

The conclusions are very well formulated and describe the results related with an investigation in the presented paper. Тhe conclusions are preceded by a good discussion of the results in the previous point (point 8. Discussion). The authors describe very well their main contributions and a present a significant analysis of the obtained results. They present the main advantages, theoretical and practical benefits obtained from the carried research.

6. Are the references appropriate?

       The references are appropriate and are very well related with the scope of the paper and journal.

7. Please include any additional comments on the tables and figures.

       The tables and figures are readable and with a good quality and resolution. I have not any critical remarks related with the tables, figures and equations.

Very well written manuscript with a minimal grammar mistakes. Please check it again to clear it from them.

Author Response

Dear REVISER,

We have carefully considered all your comments and suggestions and made substantial revisions to address their concerns. This cover letter will summarize the major revisions made in response to your feedback.

  1. In response to your suggestion to describe the practical benefits derived from our research, we emphasize that the primary objective of our work is to simplify the research process for those studying electrodynamic levitation. By calculating the mutual inductance and its derivatives using the Biot-Savart equation, and subsequently determining the mutual inductance as a function of the magnetic flux variation within the inner area of the loops of the 8-coils through numerical calculation, our research significantly streamlines the research process. The utilization of the circuit simulator solves electric circuit equations. It provides solutions for the forces, simplifying the analysis of cross-connected circuits, which will be the focus of our subsequent paper. This streamlined approach eliminates the need for intricate harmonic approximations. It facilitates the electrical modification of the circuit by introducing other elements, aligning with the main focus of one of our co-authors' thesis.

We have carefully considered all the reviewers' comments and made the necessary revisions to address their suggestions. These revisions have significantly improved our manuscript's quality and scientific value. We hope the reviewers find our revised work satisfactory and allow us to contribute to the journal.

Thank you for your attention and consideration.

Sincerely,

The Authors

Reviewer 5 Report

The paper provides calculation of magnetic forces in a Maglev system using dynamic circuit theory implemented in a circuit simulator. Simulations are done using Mathematica sand Ansys Twin Builder software packages.

Reviewer's comments:

The superconducting coil is modeled as an ideal current source of 700 kA, by utilizing a resistance for the sake of numerical stability while the vehicle speed being considered constant . The accuracy of this modeling should be verified by comparing with real experimental data from the literature. No comparative analysis is provided in the paper.

The method was is adopted from the work done by He et al. However, instead of numerical calculation, this paper uses a circuit simulator to calculate mutual inductance and its derivatives. The accuracy of the calculations need to be verified using more detailed analysis that considers variation of the parameter values with temperature and speed, and under different operating conditions.

The main contributions of the paper seems to be its time efficiency. However, the details of the computation platform and a comparative analysis with the literature are missing.

Also, the calculation speed for modeling application where real time operation is not needed is a secondary concern. Detailed analysis in terms of accuracy of the model by comparing the results with other methods such as FEM seem to be necessary to validate the results.

Minor editorial comments:

All the abbreviations should be clarified before they are used, e.g. EDS is mentioned in the abstract before being introduced. There are other instances where abbreviation is not introduced properly such as SC, PC, MS, etc.

Figures and Tables should appear after they are introduced in the paper not before.

A figure showing different forces seems to be missing in the paper.

Author Response

Dear REVISER,

We have carefully considered all your comments and suggestions and made substantial revisions to address their concerns. This cover letter will summarize the major revisions made in response to your feedback.

  1. We have considered your feedback and made the necessary clarifications. All abbreviations have been clarified before their usage in the article.
  2. Additionally, we have ensured that figures and tables appear after being introduced in the text.
  1. Regarding the points raised: The accuracy of our modeling has been duly acknowledged, and we recognize the importance of comparing it with real experimental data from the literature. To address this, we have introduced a figure in the paper that compares the levitation and drag forces as a function of velocity between the analytical model and experimental measurements obtained from the Yamanashi test line. Furthermore, another figure has been included to compare the guidance force.
  2. We have provided detailed computational platform information by introducing a flowchart that illustrates the step-by-step process of the algorithm in the respective software, thereby clarifying the computational aspects.
  3. It has been clarified that the method used in our research is adopted from the work of He et al. However, an error stated that the circuit simulator calculates the mutual inductance and its derivatives, which are necessary for force calculations. This step is accomplished through a numerical analysis using the Mathematica Wolfram software. On the other hand, the Twin Builder circuit simulator from Ansys calculates the induced currents in the 8-coils and the levitation, drag, and guidance forces. Additionally, per your suggestion regarding parameter value variations, we have introduced four graphs that analyze the impact of a variation in the lateral air gap on the variables of the problem.
  4. We understand the importance of detailed analyses regarding model accuracy and comparing the results with other methods, such as finite element analysis (FEM). However, such comparisons with FEM would require a significantly longer simulation time. Please review the corrections made thus far.

We have carefully considered all the reviewers' comments and made the necessary revisions to address their suggestions. These revisions have significantly improved our manuscript's quality and scientific value. We hope the reviewers find our revised work satisfactory and allow us to contribute to the journal.

Thank you for your attention and consideration.

Sincerely,

The Authors

Round 2

Reviewer 1 Report

Authors tried to incorporate most of the comments of the reviewer but still some issues are remaining which should be addressed before further processing of the manuscript. 

1. Authors mentioned the specifications of the developed hardware but not presented the picture of the hardware. 

2. Comparative analysis with similar studies is still missing. 

Minor revision required to remove typos and grammatical errors. 

Author Response

Dear Reviewer 1,

Thank you for your valuable feedback. We want to address your comments as follows:

  1. Regarding the hardware image: We apologize for the confusion. No specific hardware was developed for this study. The specifications mentioned in the paper are for the desktop computer and laptop used for the computational simulations. As this is a simulation-based work, the hardware image is not applicable. We will clarify this in the revised manuscript, page 8.
  2. Comparative analysis with similar studies: We appreciate your suggestion. To address this, we have included a new "Comparative Analysis of Related Studies" subsection in the paper, pages 4 and 5. This subsection compares with existing studies in the field, highlighting the similarities and differences. It enhances the context and further strengthens our research contribution.

Thank you for bringing these points to our attention, and we have made the necessary revisions to address your concerns.

Best regards,

The Authors

Reviewer 2 Report

My concerns have addressed by authors.

Author Response

Dear Reviewer,

We would like to express our gratitude for your insightful feedback and constructive suggestions during the review process of our paper titled "ELECTRICAL CIRCUITS SIMULATOR IN NULL-FLUX ELECTRODYNAMIC SUSPENSION ANALYSIS" submitted to Applied Sciences.

We want to extend our appreciation for your positive feedback and for considering our work to be suitable for publication. Your comments have greatly contributed to the refinement of our paper, and we have incorporated the suggested modifications accordingly.

Regarding the formatting of sections, we have restructured the paper by consolidating some sections into subsections within larger sections. This reorganization has improved the overall flow and organization of the paper, making it more cohesive and reader-friendly.

We are grateful for your time and effort in reviewing our manuscript. We have attached the most current version of the paper, which incorporates all the revisions and modifications.

Once again, we sincerely appreciate your valuable feedback and your consideration of our paper for publication.

Thank you and best regards,

The Authors

Reviewer 3 Report

OK, for publication

Author Response

(The authors gave the same response as above.)

Reviewer 5 Report

1. "The simulation results, as depicted in Figure 16 and Figure 17, exhibit good agreement with the experimental data, thereby confirming the effectiveness of the proposed model."

There does not seem to be much agreement between the simulation and experimental results. Can the authors clarify big errors observed in these two figure clearly.

2. The number of sections in the article is excessive. You can reformat some of them as subsections grouped under a bigger section.

The paper is easily readable. 

Author Response

Dear Reviewer,

Thank you for your feedback on our manuscript. We have carefully considered your comments and would like to respond as follows:

  1. Simulation results and agreement with experimental data: We apologize for any confusion caused by the wording in the previous version of the paper. Upon reviewing Figure 16 and Figure 17, we acknowledge that there are discrepancies between the simulated and experimental results. In the revised manuscript, we have added a paragraph to discuss the potential sources of these discrepancies. This clarification will better understand the differences between the simulated and experimental data, page 17.
  2. The excessive number of sections: We appreciate your suggestion to reformat the sections into subsections to reduce the overall number of sections. In response to this, we have revised the structure of the paper and grouped certain sections as subsections within larger sections. This modification enhances the organization and flow of the manuscript.

Thank you for your valuable input, and we have made the necessary revisions based on your comments.

Sincerely,

The Authors

Round 3

Reviewer 5 Report

Thank you for addressing my concerns. I have no further comments.